# In-Situ Grown NiMn_2_O_4_/GO Nanocomposite Material on Nickel Foam Surface by Microwave-Assisted Hydrothermal Method and Used as Supercapacitor Electrode

**DOI:** 10.3390/nano13172487

**Published:** 2023-09-04

**Authors:** Shusen Wang, Xiaomei Du, Sen Liu, Yingqing Fu, Naibao Huang

**Affiliations:** 1Department of Public Security Management, LiaoNing Police College, Dalian 116036, China; 2Materials Sciences & Engineering, Dalian Maritime University, Dalian 116026, China; missgalaxy121@outlook.com (X.D.); 1_sam@foxmail.com (S.L.); fuyingqing@dlmu.edu.cn (Y.F.); nbhuang@dlmu.edu.cn (N.H.)

**Keywords:** supercapacitor, NiMn_2_O_4_, NiMn_2_O_4_/GO, microwave-assisted hydrothermal method

## Abstract

The NiMn_2_O_4_/graphene oxide (GO) nanocomposite material was in situ grown on the surface of a nickel foam 3D skeleton by combining the solvent method with the microwave-assisted hydrothermal method and annealing; then, its performance was investigated as a superior supercapacitor electrode material. When nickel foam was soaked in GO aqueous or treated in nickel ion and manganese ion solution by the microwave-assisted hydrothermal method and annealing, gauze GO film or flower-spherical NiMn_2_O_4_ was formed on the nickel foam surface. If the two processes were combined in a different order, the final products on the nickel surface had a remarkably different morphology and phase structure. When GO film was first formed, the final products on the nickel surface were the composite of NiO and Mn_3_O_4_, while NiMn_2_O_4_/GO nanocomposite material can be obtained if NiMn_2_O_4_ was first formed (immersed in 2.5 mg/L GO solution). In a 6M KOH solution, the specific capacitance of the latter reached 700 F/g at 1 A/g which was superior to that of the former (only 35 F/g). However, the latter’s specific capacitance was still inferior to that of in-situ grown NiMn_2_O_4_ on nickel foam (802 F/g). Though the gauze-formed GO film, almost covering the preformed flower-spherical NiMn_2_O_4_, can also contribute a certain specific capacitance, it also restricted the electrolyte diffusion and contact with NiMn_2_O_4_, accounting for the performance decrease of the NiMn_2_O_4_/GO nanocomposite. A convenient method was raised to fabricate the nanocomposite of carbon and double metal oxides.

## 1. Introduction

Increasingly serious environmental problems and energy demand encourage people to study safe and efficient energy. As a new type of green energy storage device, the supercapacitor has a broad application prospect and is worth further study [1,2,3]. The performance and efficiency of supercapacitors depend directly on the electrode materials [4,5,6]. Metal oxides have been widely used as electrode materials in supercapacitors because of their high theoretical specific capacity, especially multi-metal oxides which attract more researchers’ attention because of the synergistic effect of multivalent states and multi-metal ions [7]. For example, Li et al. synthesized hierarchical MnCo_2_O_4_ nanosheets by using a two-step hydrothermal method and post-annealing treatment [8]. Its porous structure and large specific surface area have a positive effect on electrochemical activity and enhance electron diffusion. Chen et al. prepared nickel-cobalt hydroxide by electrochemical deposition on a nickel foam substrate and then annealed at 300 °C to transform the coating into porous NiCo_2_O_4_ nanosheets, which can be directly used as binderless electrodes [9]. The high specific capacitance of the electrode is 1734.9 and 1201.8 F/g at the current densities of 2 A/g and 50 A/g, respectively, indicating good magnification performance. After 3500 cycles at 30 A/g, the capacitance only decreased by about 12.7%, showing good cycle stability.

Nickel manganate (NiMn_2_O_4_) is a kind of AB_2_O_4_ spinel transition metal oxide with a stable structure and easy preparation. It is the most widely used material in negative temperature coefficient thermistors [10,11]. However, due to its relatively poor conductivity and low actual specific capacitance, NiMn_2_O_4_ is still very limited in its application as electrode material for supercapacitors. A lot of research is needed [5,12]. At present, a common means to improve the performance of metal oxide electrode materials is to compound them with carbon materials, which can solve the problem of poor conductivity of metal oxides and facilitate electron transmission [13]. Among the carbon materials, graphene, a two-dimensional carbon material, has many advantages, such as high electrical conductivity, good flexibility, and high mechanical strength, and is an ideal choice for preparing composite electrode materials.

In this work, NiMn_2_O_4_/GO composites were in situ grown on the surface of 3D skeleton foam nickel by the microwave hydrothermal method, which can be used as the binder-free electrode of the supercapacitor. During synthesis, the process was conducted according to the following different processes, respectively. One process is first depositing a layer of GO on the surface of the foamed nickel 3D skeleton, and then NiMn_2_O_4_ is grown on it. The other process is that NiMn_2_O_4_ is first grown on the surface of foamed nickel, and then coated GO. The effect of the concentration of GO aqueous solution on the performance of the obtained composite materials was studied. At 1 A/g, the specific capacitance of the nanocomposite of NiO and Mn_3_O_4_ (after immersing in 2.5 mg/L GO solution) was only 35 F/g. And the specific capacitance of NiMn_2_O_4_/GO nanocomposite material reached 700 F/g.

## 2. Experimental

### 2.1. The Synthesis of Nanocomposite

NiMn_2_O_4_/graphene oxide (GO) composite material was prepared according to the two different processes, which are illustrated in Figure 1.

The first detailed process was as follows:(1)A total of 30 mL of GO aqueous solutions with different concentrations (0, 2.5, 5, and 7.5 mg/mL) were prepared and poured into Petri dishes, respectively.(2)The pre-treated nickel foam was soaked in the petri dish for 1.5 h and then washed gently in deionized water several times, and then put into a drying oven at 60 °C for 12 h to get graphene-coated nickel foam.(3)Accurately weighted 0.5 mmol nickel nitrate hexahydrate, 3 mmol ammonium fluoride, and 7.5 mmol urea were all dissolved in 30 mL deionized water and magnetically stirred for 15 min to obtain a clear, light-green solution. Then, 1 mmol potassium permanganate was added into the above solution, which was continuously magnetically stirred for 30 min. Finally, the purplish-red solution was obtained.(4)The treated foam nickel was put into a 100 mL microwave reactor containing the purplish-red solution and then put into the microwave synthesis instrument. The reactor was heated to 140 °C for 3 h and then cooled to room temperature.(5)The nickel foam was taken out, ultrasonicated with deionized water for 5 min and anhydrous ethanol several times, and then dried in a drying oven at 60 °C for 12 h.(6)The dried nickel foam was annealed at 450 °C for 2 h in a tube furnace with a heating rate of 10 °C/min at N_2_ atmosphere and then cooled to room temperature in the furnace.

The second detailed process was as follows:(1)Accurately weighted 0.5 mmol nickel nitrate hexahydrate, 3 mmol ammonium fluoride, and 7.5 mmol urea were all dissolved in 30 mL deionized water, and magnetically stirred for 15 min to obtain a clear, light-green solution. Then, 1 mmol potassium permanganate was added into the above solution and magnetically stirred for 30 min. The purplish-red solution was obtained.(2)The pre-treated nickel foam was immersed into a 100 mL microwave container having the prepared solution for 30 min. The container was put into the microwave synthesis instrument and heated to 140 °C for 3 h with certain procedures and then cooled to room temperature.(3)The nickel foam was removed, cleaned by ultrasonic for 5 min with deionized water and anhydrous ethanol several times, and then dried in a drying oven at 60 °C for 12 h.(4)The dried nickel foam was heated to 450 °C with a heating rate of 10 °C/min in a tube furnace at N_2_ atmosphere held for 2 h and then cooled to room temperature. The nickel foam with NiMn_2_O_4_ on the surface was obtained.(5)The treated nickel foam was put in a Petri dish containing 30 mL of GO aqueous solution with different concentrations and immersed for 1.5 h.(6)The soaked nickel foam was gently washed with deionized water several times, and then dried in a blast oven at 60 °C for 12 h.

All the reagents (see Table 1) were of analytical grade and used without further purification.

### 2.2. Characterization

The obtained samples were characterized by X-ray diffraction (XRD: D/MAX-Ultima+, Co Ka radiation; 8 °C/min), the field-emission scanning electron microscope (FESEM: Supra-55-sapphire, Zeiss, Oberkochen, Germany), and X-ray photoelectron spectroscopy (XPS) were conducted on the T64000 system and ESCALAB250Xi, respectively.

### 2.3. Electrochemical Tests

Electrochemical tests including cyclic voltammetry (CV) and galvanostatic charge/discharge (GCD) were conducted on a VMP3 (EG&G) electrochemical workstation in 6M KOH solution using a three-electrode system, i.e., the treated electrode used as the working electrode, Platinum net as the counter electrode, and the saturated calomel electrode (SCE) as the reference electrode, respectively. CV curves were recorded in the potential range of −0.15 V to 0.55 V with different scan rates (10, 20, 50, and 100 mV/s). The galvanostatic charge-discharge (GCD) was performed at a current density of 1 A/g between about −0.15 and 0.55 V. The specific capacitance calculation formula based on the GCD curve is as follows [14]:(1)$$C_{m}=\frac{i\times ∆t}{m\times ∆V}=\frac{i\times ∆t}{S\times ∆V}$$
where *C_m_* represents the specific capacitance of the electrode material; the unit is F/g or F/cm^2^; *i* is the discharge current; the unit is A; *t* represents the discharge time; the unit is s; *m* is the mass of the active substance in g; *S* is the electrode area; the unit is cm^2^; *V* represents the discharge voltage; the unit is V. The capacity of the NiMn_2_O_4_ is calculated according to the mass change of the nickel foam before and after being coated with NiMn_2_O_4_. This sentence is also added to the experiment.

## 3. Results and Discussion

### 3.1. The Characteristic of Composite Material Fabricated According to the First Process

Figure 2 illustrates the FE-SEM images of nickel foam before and after being coated with GO by the first process. As shown in Figure 2a,b (a—pre-treated nickel foam, b—after soaked in 2.5 mg/L GO), compared with pretreated nickel foam, it seems that a wrinkle film was formed after immersion in the solution having GO, which may prove that the 3D skeleton surface of nickel foam was successfully coated with GO in profile. From Figure 1c, the uniformly and tightly film was formed on the surface of nickel foam after the microwave hydrothermal method and annealing. After the nickel foam soaked in solution, having a different content of GO (d~f), it was clear that deposits were observed. Meanwhile, the higher the concentration of GO, the thicker the obtained film, and the more serious the film fell off.

Figure 3 illustrates the XRD pattern of the microwave hydrothermal method and annealed nickel foam with and without soaking in 2.5 mg/L GO solution. As shown in Figure 3a, for nickel foam without immersion in GO solution, its diffraction peak occurred at 21.26°, 35.06°, 41.41°, 43.57°, 50.68°, 67.50°, and 74.37°, corresponding to the (111), (220), (311), (222), (400), (511), and (440) plane of the standard of cubic NiMn_2_O_4_ (JCPDS#01-1110). The result indicated that NiMn_2_O_4_ was formed on the surface of the microwave hydrothermal and annealed nickel foam without soaking in GO solution. But for nickel foam immersed in 2.5 mg/L GO solution, the diffraction peak was composed of the characteristic peaks of NiO (JCPDS#47-1049) and Mn_3_O_4_ (JCPDS#80-0382). The preferential-formed GO film on the nickel foam surface may participate in the redox reaction during the microwave hydrothermal process, which led to the phase change of microwave hydrothermal products. As a result, NiMn_2_O_4_ was not obtained [15,16].

The CV curves of the obtained samples measured at different scan rates (10, 20, 50, and 100 mV/s, respectively) were summarized in Figure 4. From CV curves, for all the obtained samples, the current increased gradually with the increase in the scan rate, while its shape changed little, indicating that the electrochemical behavior of the sample will be slightly affected by the polarization [17]. From Figure 4a, an obvious redox peak was observed because of the existence of NiMn_2_O_4_ on the surface of nickel foam, which exhibited pseudo-capacitance behavior. In Figure 4b–d, the CV curve of the obtained sample was obviously different from that of Figure 4a with the redox peak having a great change. It can be found that at the same scan rate, the absolute integral area of the CV curve of the sample immersed in GO aqueous solution was much smaller than that of pure NiMn_2_O_4_ (Figure 4a), meaning the performed GO coating had no benefit for improving its specific capacitance. Since nickel foam was first immersed in aqueous solution having GO, its surface was curved by GO film (Figure 2b), which may affect the formation of NiMn_2_O_4_ during the subsequent microwave hydrothermal and anneal process, confirmed by the result of Figure 3b. So, the change in CV curves related to the formed GO film and these oxides.

Figure 5a illustrates the GCD curves of the obtained samples according to the first process at 1 A/g between the potential ranges of −0.15~0.5 V. On the one hand, the GCD curve of annealed nickel foam with (2.5, 5.0, 7.5 mg/L) GO solution presents a nearly symmetrical shape, and the curve shape is basically similar under different current densities. The results show that the electrode has high coulomb efficiency and excellent reversibility. On the other hand, it is obvious that the charge and discharge time of the sample without immersing in GO aqueous solution were much longer than those samples soaking in GO solution, representing pseudo-capacitance characteristics. Meanwhile, the calculated specific capacitance based on Equation (1) was summarized in Figure 5b. It can be clearly seen that all the obtained samples after immersing in GO solution had a very small specific capacitance, which was significantly inferior to the sample without soaking in GO solution. To be specific, at 1 A/g the electrode impregnated at 2.5 mg/L GO solution has a maximum specific capacitance of 493.7 F/g, which is higher than that of the electrode impregnated at 5 mg/L and 7.5 mg/L GO solution. It is worth noting that when the electrode does not invade the bubble, the GO solution ratio is surprisingly high for 802 F/g. This result corresponded to that of the CV curve.

### 3.2. The Performance of NiMn_2_O_4_/GO Nanocomposite Material

The micro-morphology of the samples prepared according to the second process is shown in Figure 6. From Figure 6a, the surface of nickel foam without immersing in having the GO solution was covered by a bouquet assembled with nanosheet, which was NiMn_2_O_4_ (as shown in Figure 3a). As shown in Figure 6b–d, it can be clearly seen that the bouquet was covered by thin gauze film, and the thickness of the gauze film increased with the GO concentration increasing.

Figure 7 illustrates the composition analysis result from EDS of the obtained sample soaked in 2.5 mg/L GO solution. It can be seen that the sample was composed of Ni, Mn, O, and C elements. Meanwhile, these elements were all uniformly scattered on the surface of nickel foam. The results combined with that of Figure 6 further confirmed that the previously formed NiMn_2_O_4_ on the surface of nickel foam was successfully coated by gauze GO.

In order to determine the elemental composition and chemical valence of the prepared samples further, XPS analysis of the samples after soaking in 2.5 mg/mL GO aqueous solution was performed (see Figure 8). The total spectrum of the sample (Figure 8a) identified Ni, Mn, O, and C element peaks, which was consistent with the composition. The Ni 2p spectrum (Figure 8b) was well deconvoluted into two Ni^2+^ fitted peaks at 855.7 eV and 873.5 eV, two Ni^3+^ fitted peaks at 861.6 eV and 879.9 eV [18], and two shakeup satellites. The fitting results of Mn 2p in Figure 7c indicated the existence of Mn 2p1/2 and Mn 2p3/2. The fitting peaks of Mn 2p1/2 and Mn 2p3/2 revealed the coexisting of Mn^2+^ and Mn^3+^ oxidation states in the product, which occurred at 640.9 eV, 651.6 eV, 642.5 eV, and 653.0 eV [19], respectively. The O 1s spectrum in Figure 8d could be deconvoluted into lattice oxygen (M-O) at 530.4 eV, metal-O-H at 531.8 eV, and surface oxygen species (H-O-H) at 532.9 eV [13,20], respectively. Figure 8e showed the C 1s spectrum, with three peaks at 284.4 eV, 285.1 eV, and 286.9 eV, corresponding to the C=C bond, C-O bond, and C=O bond [21], respectively. The XPS results further confirm the successful in-situ growth of NiMn_2_O_4_/GO nanocomposite materials on nickel foam.

Figure 9 displayed the CV curves of obtained samples according to the second process (soaking in 0, 2.5, 5.0, and 7.5 mg/L GO solution) at different scan rates (10, 20, 50 and 100 mV/s), respectively. It is easy to see that all CV curves have obvious redox peaks, indicating that all samples behaved with pseudo-capacitance behavior. As shown in Figure 9, the CV curve of the sample having GO coating was obviously different from that of the sample without GO coatings. The difference was shown in the positions of the redox peak. That is, the redox potential of the latter was earlier than that of the former, indicating that the formed GO coating had an influence on its electrochemical behavior. In addition, it can be found that the current gradually increased with the scan rate increasing, while its shape changed little, indicating that the electrochemical behavior of the sample was less affected by the polarization. However, for the sample obtained after being soaked in 7.5 mg/L GO solution (Figure 9d), the redox peak on CV was deformed, illustrating that the electrochemical behavior of the sample was more susceptible to polarization when the GO coating was too thick.

Figure 10a displays the GCD curves of the obtained sample after being soaked in 2.5 mg/L GO aqueous solutions at a current density of 1 A/g between −0.2 to 0.5 V. The charge-discharge platforms occurred on all the GCD curves, representing the characteristics of pseudocapacitors. From the charge and discharge time point of view, the time of the sample without GO film was longer than that of those having GO coatings. The calculated corresponding specific capacitance by Equation (1) based on the GCD curve at different current densities is shown in Figure 10b. At different current densities of 1–10 A/g, there is no significant difference in the specific capacitance of the impregnated 2.5, 5, and 7.5 mg/mL GO film electrodes, especially at high current densities. Obviously, the specific capacitance of samples having GO film had little difference and was all smaller than that of samples without GO coating. Combined with the FE-SEM analysis results, it can be seen that the preformed flower-spherical-like NiMn_2_O_4_ was almost completely covered by the gauze GO film. Meanwhile, the larger the GO concentration, the thicker the gauze. Although the GO film also can contribute a certain specific capacitance, the preformed NiMn_2_O_4_ was covered by GO film, blocking electrolyte diffusion and contact. As a result, its specific capacitance was worse.

## 4. Conclusions

The nanocomposite was in situ grown on the surface of nickel foam by combining the solvent method with microwave-assisted synthesis followed by heating and annealing according to a different order. Meanwhile, the morphology, structure, and electrochemical performance of the obtained nanocomposite were investigated. When a nickel foam was first immersed in having GO aqueous solution and then treated in the solution containing nickel nitrate hexahydrate, fluoride, urea, and potassium permanganate solution by microwave-assisted synthesis and heating annealing process, the nanocomposite of NiO and Mn_3_O_4_ was formed on nickel foam in-situ. At 1 A/g, the specific capacitance of the nanocomposite of NiO and Mn_3_O_4_ (after immersing in 2.5 mg/L GO solution) was only 35 F/g. While the process was reversed, the NiMn_2_O_4_/GO nanocomposite material can be directly grown on the surface of a foamed nickel 3D skeleton. At 1 A/g, the specific capacitance of NiMn_2_O_4_/GO nanocomposite material reached 700 F/g. Overall, this synthesis strategy provides a convenient method for preparing carbon and bimetallic oxide nanocomposites. In addition, this study can open up opportunities for the application of these nanocomposites in supercapacitors.

## Figures and Tables

**Figure 1 nanomaterials-13-02487-f001:**
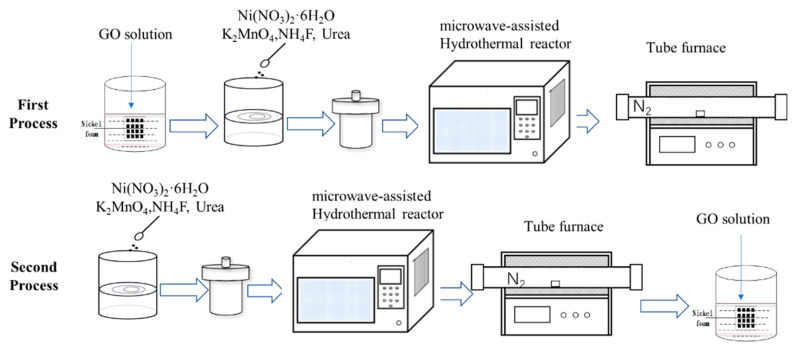
The illustration of synthesis procedure.

**Figure 2 nanomaterials-13-02487-f002:**
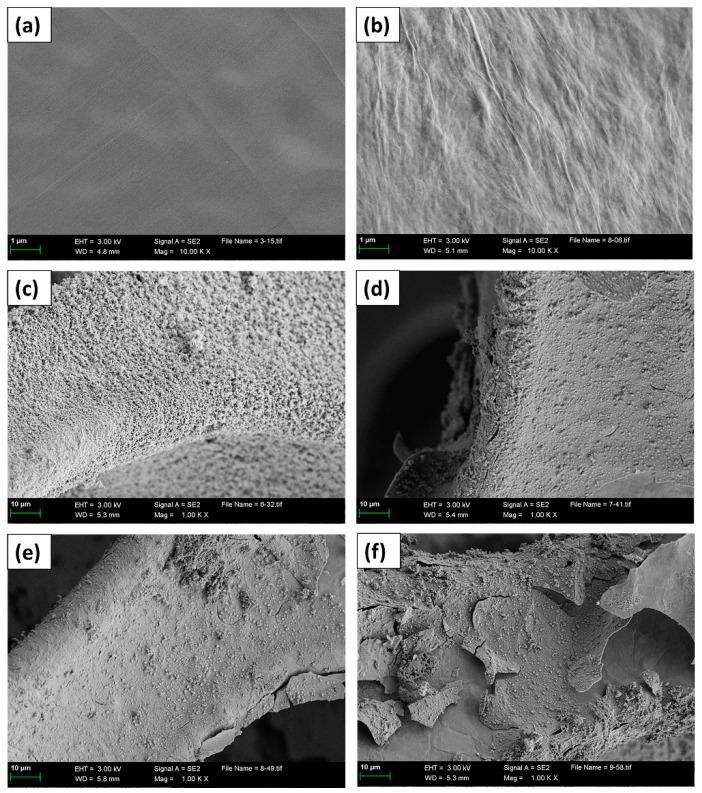
FE-SEM images of the obtained samples (**a**) Pickling Nickel foam, (**b**) dried pickling nickel foam after soaked in 2.5 mg/L GO solution, (**c**–**f**) microwave hydrothermal and annealed pickling nickel foam after soaked in different GO solution (0, 2.5, 5.0, and (**f**)—7.5 mg/mL).

**Figure 3 nanomaterials-13-02487-f003:**
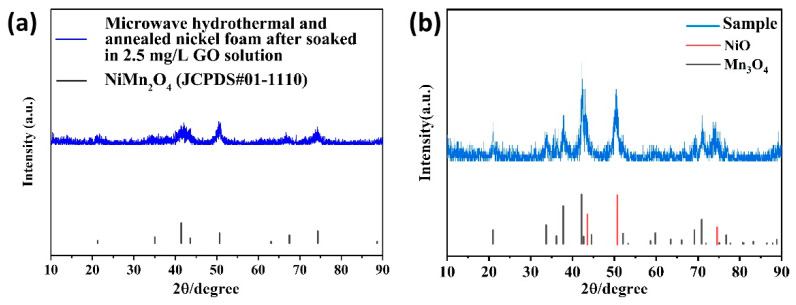
XRD pattern of microwave hydrothermal and annealed nickel foam without (**a**) and with (**b**) soaking in 2.5 mg/L GO solution.

**Figure 4 nanomaterials-13-02487-f004:**
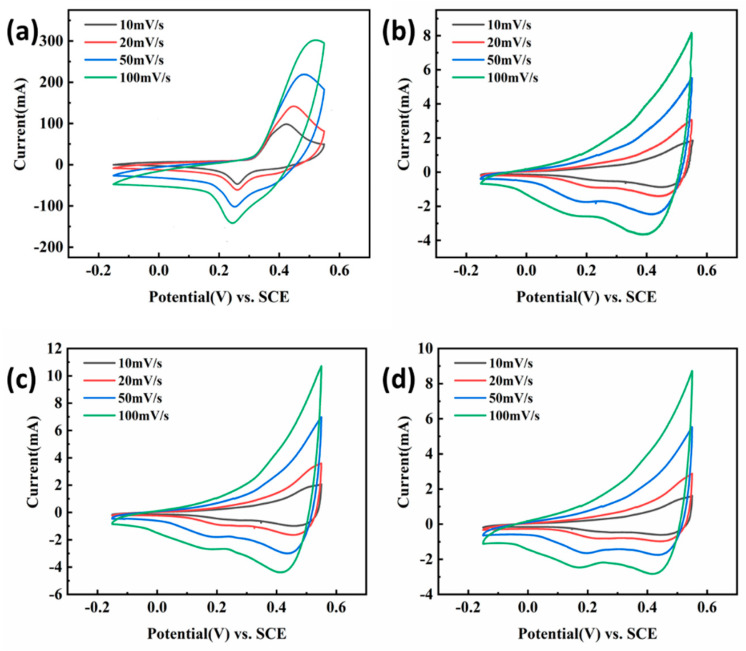
CV curves of microwave hydrothermal and annealed nickel foam without (**a**) and with ((**b**)—2.5, (**c**)—5.0, (**d**)—7.5 mg/L) GO solution in 6M KOH solution.

**Figure 5 nanomaterials-13-02487-f005:**
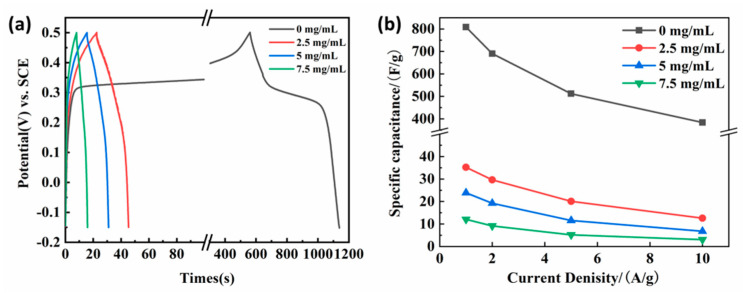
(**a**) GCD curves at 1 A/g and (**b**) the calculated specific capacitance at different current densities (1, 2, 5, and 10 A/g) of the hydrothermal and annealed nickel foam without and with (2.5, 5.0, 7.5 mg/L) GO solution in 6M KOH solution.

**Figure 6 nanomaterials-13-02487-f006:**
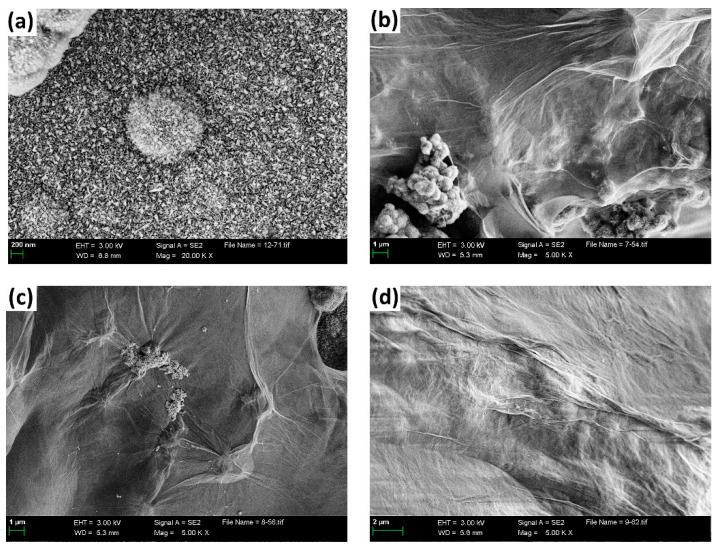
FE-SEM images of the samples immersing in (**a**) 0 mg/L, (**b**) 2.5 mg/L, (**c**) 5.0 mg/L, and (**d**) 7.5 mg/L GO solution after hydrothermal and anneal.

**Figure 7 nanomaterials-13-02487-f007:**
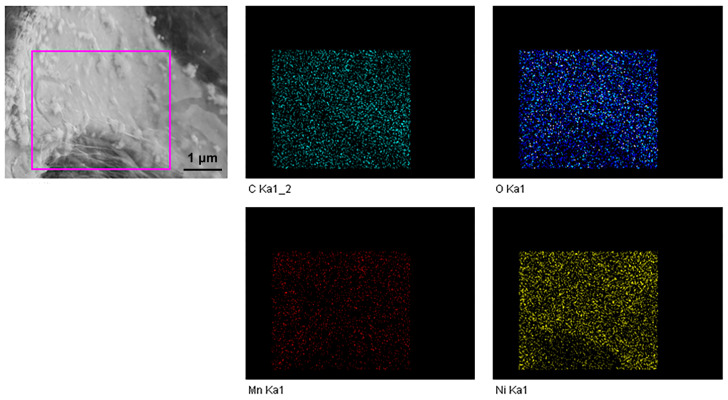
EDS images of the obtained sample after immersing in 2.5 mg/mL GO solution.

**Figure 8 nanomaterials-13-02487-f008:**
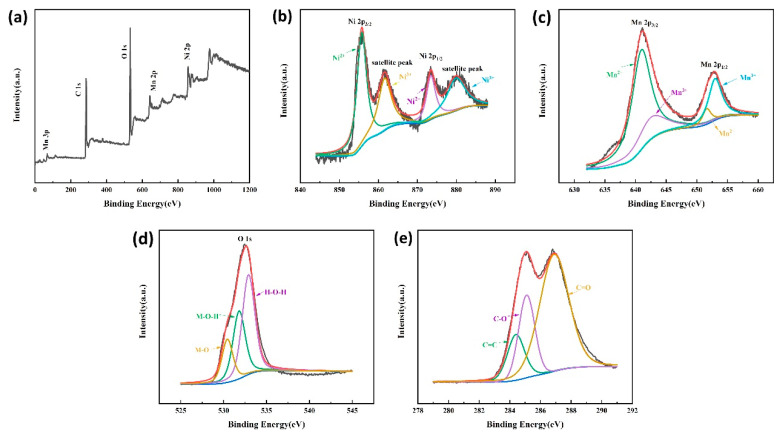
The XPS spectra of the obtained sample after immersing in 2.5 mg/mL GO solution (**a**) a total survey (**b**) Ni 2p (**c**) Mn 2p (**d**) O 1s (**e**) C 1s.

**Figure 9 nanomaterials-13-02487-f009:**
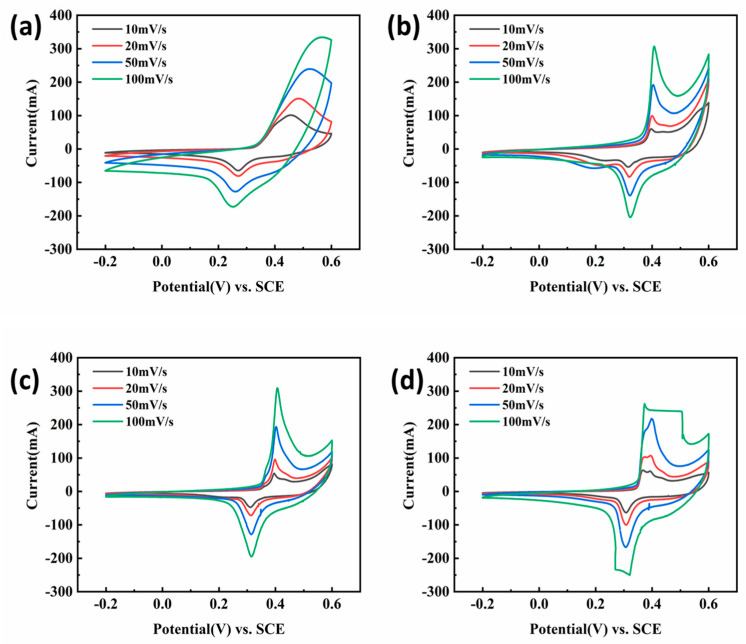
The CV curves of the obtained samples in 6M KOH solution after immersing in (**a**) 0 mg/mL, (**b**) 2.5 mg/mL, (**c**) 5 mg/mL, and (**d**) 7.5 mg/mL GO aqueous solutions.

**Figure 10 nanomaterials-13-02487-f010:**
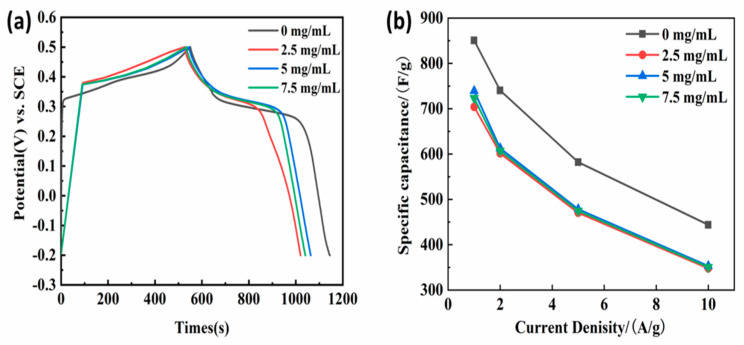
(**a**) GCD curves at 1 A/g and (**b**) the calculated specific capacitance at different current densities (1, 2, 5, and 10 A/g) of the obtained sample without and with GO film in 6M KOH solution.

**Table 1 nanomaterials-13-02487-t001:** Reagents for experiment.

Name	Grade	Manufacturers
Nickel dinitrate hexahydrate	analytical	Tianjin Damao Chemical Reagent Factory (Tianjin, China)
Potassium permanganate	analytical	Tianjin Kemio Chemical Reagent Co., Ltd. (Tianjin, China)
urea	analytical	Fucheng (Tianjin) Chemical Reagent Co., Ltd. (Tianjin, China)
ammonium fluoride	analytical	Tianjin Kemio Chemical Reagent Co., Ltd. (Tianjin, China)
hydrochloric acid	analytical	Shenyang Liansheng Chemical Co., Ltd. (Shenyang, China)
N_2_	99.99%	Dalian bright Special Gas Co., Ltd. (Dalian, China)
anhydrous ethanol	analytical	Tianjin Fuyu Fine Chemical Co., Ltd. (Tianjin, China)
Nickel foam	----------	Ai Lantian HIGH-TECH Materials (Dalian) Co., Ltd. (Dalian, China)
graphene oxide solution	10 mg/mL 5 mg/mL	Suzhou Hengqiu Technology Co., Ltd. (Suzhou, China)

## Data Availability

Not applicable.

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
