# Peer review of "In-Situ Grown NiMn2O4/GO Nanocomposite Material on Nickel Foam Surface by Microwave-Assisted Hydrothermal Method and Used as Supercapacitor Electrode"

_nanomaterials, 2023, doi:10.3390/nano13172487_

Round 1

Reviewer 1 Report

Many more articles on NiMn2O4 materials for capacitors and pseudo capacitors than those cited in the draft exist in the literature. It would be good to have a comparison table of where the submitted work sits with respect to prior art.

The english language needs improvements. I have made some annotations where the language needs to be checked and corrected.

Author Response

[Q1] Many more articles on NiMn2O4 materials for capacitors and pseudo capacitors than those cited in the draft exist in the literature. It would be good to have a comparison table of where the submitted work sits with respect to prior art.

[R1] Thanks for the reviewer’s comments. Since the specific capacitance of the nanocomposite was inferior to that of the nickel foam coated with NiMn2O4, it was unnecessary to compare with the previously reported materials.

[Q2] The english language needs improvements. I have made some annotations where the language needs to be checked and corrected.

[R2] We appreciate the reviewer for pointing out this issue. 

The manuscript has been carefully checked and revised by the experts in this field.

 Other Comments and Suggestions  "Please see the attachment."

Reviewer 2 Report

In the present report, the authors constructed the NiMn2O4/graphene oxide (GO) nanocomposite material using in-situ grown on the surface of nickel foam 3D skeleton by combining solvent method with microwave-assisted hydrothermal method and annealing, then its performance was investigated by electrochemical techniques. In 6M KOH solution, the specific capacitance of the latter reached 700 F/g at 1 A/g which was more superior to that of the former (only 35 F/g). But the latter’s specific capacitance was still inferior to that of in-situ grown NiMn2O4 on nickel foam (802 F/g). This paper provided some valuable information and the content is very significant in this field. However, I recommended a major revision of the article from its present form before it can be published in ijms. Some specific comments are as follows:

1.         The abstract and conclusion sections should be a specific and scientific approach.

2.          In the introduction section, the authors should expound the research significance of the present work.

3.         The authors should explain the novelty of the present report?

4.         The authors should provide a clear schematic representation of the formation mechanism.

5.         What is the pH of the reaction solution? The pH of the solution normally varies from precursor to precursor. The authors must justify the selection of pH, temperature and time.

6.         Why was this method chosen? Are any other methods used for the synthesis of NiMn2O4/ graphene oxide (GO) composite?

7.         Compare with only coated NiMn2O4 and GO layer.

8.         The overall work is not well organized. Author can improve by referring to the paper: Porous Ternary High Performance Supercapacitor Electrode Based on Reduced Graphene Oxide, NiMn2O4, and Polyaniline. https://doi.org/10.1016/j.electacta.2016.09.030.

9.         Line 59: statement-In this paper, NiMn2O4/rGO composites were prepared. Everywhere it is mentioned as GO including title.

10.       Line 63: statement- During synthesis, reduced graphene oxide by two processes. It is not clear.

11.       How can you support usage of calomel electrode over Hg/HgO for KOH solution?

12.       Can you provide pictures of Ni foam coated with Active material? Also specify its weight of active material loaded on Ni foam.

13.       Why 6M KOH solution fixed for this study. Why can’t be either higher or lower?

14.       All images are very poor resolution. Authors should produce high quality images.

15.       Line 164- it is not Fig. 2, its Fig. 3. Total explanation is like that. The next GCD analysis is based on Fig.4.

16.       Why specific capacitance was calculated based on GCD curves, why not from on CV concept? What is the difference, justify.

17.       EIS studies were missing

18.       GCD long run studies were missing.

19.       Authors failed to provide a deep analysis and proper data in the submitted manuscript. Please revise the manuscript carefully.

20.       In the current state, there are more typographical errors and the language should be improved. Therefore, the authors are advised to recheck the whole manuscript for improving the language and structure carefully.

Authors failed to provide a deep analysis and proper data in the submitted manuscript, please revise the manuscript carefully.

In the current state, there are more typographical errors and the language should be improved. Therefore, the authors are advised to recheck the whole manuscript for improving the language and structure carefully.

Reviewer 3 Report

Wang et al. studied the in-situ grown NiMn2O4/GO nanocomposite material on a nickel foam surface by microwave-assisted hydrothermal method and used as a supercapacitor electrode. The manuscript should be accepted after addressing the following issues;

1)      The authors should check the title of the manuscript carefully, specifically spelling etc.

2)      In material and methods, there is no information related to the chemical source, purification etc. The authors should provide all this information in separate sections.

3)      The authors put the synthesis procedure in bullets. Its better; the authors should add a schematic diagram of the synthesis procedures.   

4)      The authors did not mention the information related to the GO, purchased or synthesized.

5)      The authors should add Raman spectroscopy of the GO and other materials

6)      To observe the actual size and lattice parameter, the authors should provide the TEM and add more information, such as BET measurement. In electrochemical measurement, BET measurement is compulsory to estimate the surface area.

7)      The main critical point is why is the GCD measured at the different potential windows and at small potential. The window of the GCD and CV is different.

8)      Whats about the specific capacity of the CVs? The authors should add-relation and this information.

9)      In GCD, what about the charging and discharging time of the device?

10)  Why the authors did not check the GCD of the other samples?

11)  The authors measure the b value of and compare it with standard values and confirm the supercapacitor behaviour

12)  The authors cited some of the latest work, such as 10.1016/j.est.2023.108022; 10.1021/acsami.5b01745

13)  What about the device's stability? Good cycling stability and rate performance are the advantages of the proposed material. To verify the high cycling performance, the structural and morphological characterizations of doped ferrites material after cycling (e.g. 1000 cycles) should be provided.

14)  The authors should measure the bare NF and the after-deposition of active materials to check the actual capacitance. 

15)  Actually, to characterize the structures of composites, it is critical to reveal the interaction/interfaces of the components because compatibility holds the key to the performance of the materials. In this regard, further efforts should be devoted.

16)  A table comparing your work with the previously reported literature should be mentioned.

17)  There are many spelling, grammatical and formatting typos in this paper.

 There are many spelling, grammatical and formatting typos in this paper.

Round 2

Reviewer 1 Report

The clarity of the article has improved substantially. When the capacity is quoted as 700F/g, does the weight include the weight of the Nickel foam? It would be good to very clearly state whether it does or it does not.Since the introduction mentions the work by Chen et al producing high specific capacity of > 1700F/g, which is much higher than the values in the current article, it would be good to emphasise the major point of difference, which would appear to be the simplicity of the process compared to that of Chen. Apart from that, the authors have that a good job addressing previous comments.

Minor editorial revision for may be required.

Author Response

[Q1]The clarity of the article has improved substantially. When the capacity is quoted as 700F/g, does the weight include the weight of the Nickel foam? It would be good to very clearly state whether it does or it does not.Since the introduction mentions the work by Chen et al producing high specific capacity of > 1700F/g, which is much higher than the values in the current article, it would be good to emphasise the major point of difference, which would appear to be the simplicity of the process compared to that of Chen. Apart from that, the authors have that a good job addressing previous comments.

[R1]Thank you for your valuable and thoughtful comments. In this work, the capacity of the NiMn2O4 is calculated according the mass change of the nickel foam before after coated with NiMn2O4. This sentence is also added to the experiment. The description of the experiment has been revised and highlighted in lines 143-145 of page 5 of the revised manuscript.

Reviewer 2 Report

The manuscript can be acceptable in the present form.

Author Response

[Q1]The manuscript can be acceptable in the present form.

[R1] Thanks for the reviewer’s comments.

Reviewer 3 Report

Accepted in the present form.

Minor editing of English language required

Author Response

[Q1] Minor editing of English language required.

[R1] We would like to thank the reviewer for this valuable suggestion. The manuscript has been carefully checked and revised by the experts in this field. The changed contents in the revised manuscript are marked in the highlighted font.